# Stigma, Sociodemographic Factors, and Clinical Factors Associated with Psychological Distress among COVID-19 Survivors during the Convalescence Period: A Multi-Centre Study in Malaysia

**DOI:** 10.3390/ijerph20053795

**Published:** 2023-02-21

**Authors:** Nur Iwana Abdul Taib, Nik Ruzyanei Nik Jaafar, Nazirah Azman, Mohammad Farris Iman Leong Bin Abdullah, Nurul Ain Mohamad Kamal, Azlin Baharudin, Muhammad Najib Bin Abdullah, Suresh Kumar Chidambaram, Alif Adlan, Loong Hui Tan, Satya Tamilselvam, Mohd Shahrir Mohamed Said, Anuar Abd Samad, Siti Nordiana Binti Dollah

**Affiliations:** 1Department of Psychiatry, Faculty of Medicine and Health Sciences, University Malaysia Sarawak, Kota Samarahan 94300, Malaysia; 2Department of Psychiatry, Faculty of Medicine, University Kebangsaan Malaysia, Kuala Lumpur 56000, Malaysia; 3Department of Community Health, Advanced Medical and Dental Institute, Universiti Sains Malaysia, Kepala Batas 13200, Malaysia; 4Department of Psychiatry & Mental Health, Hospital Sungai Buloh, Ministry of Health Malaysia, Sungai Buloh 47000, Malaysia; 5Department of Medicine, Hospital Sungai Buloh, Ministry of Health Malaysia, Sungai Buloh 47000, Malaysia; 6Department of Medicine, Faculty of Medicine, University Kebangsaan Malaysia, Kuala Lumpur 56000, Malaysia; 7Health Technology Assessment Section, Medical Development Division, Ministry of Health Malaysia, Putrajaya 62590, Malaysia; 8Department of Psychiatry, Hospital Angkatan Tentera Tuanku Mizan, Kuala Lumpur 53300, Malaysia

**Keywords:** COVID-19 survivors, psychological distress, perceived stigma, post-hospitalisation

## Abstract

High rates of psychological distress among COVID-19 survivors and stigmatisation have been reported in both early and late convalescence. This study aimed to compare the severity of psychological distress and to determine the associations among sociodemographic and clinical characteristics, stigma, and psychological distress among COVID-19 survivors across two different cohorts at two different time points. Data were collected cross-sectionally in two groups at one month and six months post-hospitalisation among COVID-19 patient from three hospitals in Malaysia. This study assessed psychological distress and the level of stigma using the Kessler Screening Scale for Psychological Distress (K6) and the Explanatory Model Interview Catalogue (EMIC) stigma scale, respectively. At one month after discharge, significantly lower psychological distress was found among retirees (*B* = −2.207, 95% confidence interval [95% CI] = −4.139 to −0.068, *p* = 0.034), those who received up to primary education (*B* = −2.474, 95% CI = −4.500 to −0.521, *p* = 0.014), and those who had an income of more than RM 10,000 per month (*B* = −1.576, 95% CI = −2.714 to −0.505, *p* = 0.006). Moreover, those with a history of psychiatric illness [one month: (*B* = 6.363, 95% CI = 2.599 to 9.676, *p* = 0.002), six months: (*B* = 2.887, CI = 0.469–6.437, *p* = 0.038)] and sought counselling services [one month: (*B* = 1.737, 95% CI = 0.385 to 3.117, *p* = 0.016), six months: (*B* = 1.480, CI = 0.173–2.618, *p* = 0.032)] had a significantly higher severity of psychological distress at one month and six months after discharge from the hospital. The perceived stigma of being infected with COVID-19 contributed to greater severity of psychological distress. (*B* = 0.197, CI = 0.089–0.300, *p* = 0.002). Different factors may affect psychological distress at different periods of convalescence after a COVID-19 infection. A persistent stigma contributed to psychological distress later in the convalescence period.

## 1. Introduction

Malaysia reported their first COVID-19 case in January 2020, which was transmitted from international travelers. In March 2020, a more massive outbreak occurred from an annual mass religious assembly in Kuala Lumpur, and the public was advised to practise social distancing. The Government of Malaysia implemented the first lockdown via a Movement Control Order (MCO) nationwide following a drastic increment in COVID-19 cases beginning on 18 March 2020. In Malaysia, moderate to very high levels of psychological distress as a result of the COVID-19 pandemic were similarly reported [1]. Anxiety, particularly, was reported to be severe among university students, Malaysian women who were Malays and pregnant suffered from a loss of income during the pandemic [2,3,4]. Another study in a neighbouring country, Myanmar, reported more psychological distress among those who were self-employed and older than 45 years old [5].

Multiple studies have shown that the rate of psychological distress among COVID-19 survivors in early convalescence was high, and a high level of post-traumatic stress was seen especially among those who were symptomatic [6,7]. A systematic review and meta-analysis found that a high prevalence of depression, anxiety, insomnia, and PTSD was documented among COVID-19 survivor regardless of gender, group, or region [8]. These psychological complications were precipitated by a lack of control among infected people, job losses, wage losses, and uncertainty about the future [9]. Both female gender and the persistence of symptoms were risk factors in developing psychological distress among COVID-19 survivors 9 months after discharge [10]. Moreover, a study among COVID-19 survivors in the Philippines found that the prevalence of anxiety and depression 8 weeks after discharge were significantly reduced [11]. However, the available evidence regarding long-term psychological distress among COVID-19 survivors is still insufficient.

It has been shown that people who were impacted financially due to COVID-19 and drank alcohol in the past four weeks were more likely to have higher levels of psychological distress [1]. A systematic review showed that, among health care workers during the pandemic, factors such as a younger age, the female gender, and a low monthly household income were associated with psychological distress [12]. Similarly, some sociodemographic factors were found to be associated with psychological distress after being infected with COVID-19, such as age, gender, employment status, and perceived stigma [13].

Stigmatisation is not uncommon during disease outbreaks. The stigma during the COVID-19 pandemic was mainly associated with those who were infected with it, those at risk of being infected, such as healthcare workers, and those from a particular race and groups that were linked to the initial spread of the illness [14]. The fear of a disease with an unknown cause may lead to stigmatisation. Among the Malaysian public, a study found that higher levels of psychological distress were associated with higher levels of fear of COVID-19 [1]. A local study among healthcare workers found that higher cautious attitudes towards COVID-19 significantly predicted higher anxiety scores [15]. This was further explored in another study which revealed that the “fear of COVID-19” and “stress of COVID-19” were associated with psychological distress among health care workers [12]. Multiple studies also concluded that the fear associated with the pandemic, containment measures, high numbers of people infected, and deaths were associated with high prevalence rates of psychological distress across populations [16,17,18]. The lack of knowledge regarding COVID-19, as it was a relatively new disease, may have had a large contribution in the development of the stigma during the pandemic. Fear of the infection among the public led to negative feelings, such as anxiety, anger, resentment, hostility, and disgust. This led to social rejection and the discrimination of people who were being labelled. Similar occurrences of stigma were reported in different countries around the world during the pandemic [19,20]. However, the data were still limited in exploring the association between stigma and psychological distress among COVID-19 survivors.

Studies have shown that more severe symptoms of anxiety and stress were reported among those with chronic diseases and a history of medical and/or psychiatric illnesses [21,22,23]. This may be due to postponement, inaccessibility to medical services and treatment, and sensitivity to external stressors associated with the pandemic [24,25]. Moreover, psychiatric ill health at follow-up was found to be associated with persistent physical symptoms, such as breathlessness and myalgia. Multiple studies reported significant psychiatric morbidity, such as anxiety and depression after 6 months of acute infection and hospitalisation for COVID-19 [26,27].

Coping style has also been reported to be associated with psychological distress during the COVID-19 pandemic. Those with a positive coping style during the pandemic may have promoted emotional well-being, whereas those with a negative coping style showed higher levels of psychological distress [28]. One strategy to cope with psychological distress is to seek psychological assistance, which can be in the form of counselling [29]. A study among college students in China during the pandemic found that those who sought counselling had higher fear, depression, and trauma scores compared to those who did not seek counselling. They also showed that experience with seeking psychological help, as well as perceived mental health, could effectively predict psychological help-seeking behaviour [29]. Most of these studies were conducted among the public; thus, data were limited regarding counselling-seeking behaviour among COVID-19 survivors.

At the time of writing, the peak of the COVID-19 pandemic had passed, and normal living had resumed in many countries. However, the recurrence of a similar pandemic in the future is possible, as many new emerging pathogens have been identified that could be a potential public health threat [30,31]. The negative impact of the COVID-19 pandemic on mental health indicated that we were not fully prepared to cope with its negative consequences. This study aimed to compare the severity of psychological distress among COVID-19-infected survivors and to determine the associations among sociodemographic and clinical characteristics, stigma, and psychological distress among survivors across two different cohorts at two different time points (one month and six months post-hospitalisation). To our current knowledge, this is the first study conducted among COVID-19 survivors in Malaysia to investigate psychological and psychosocial impact, particularly exploring stigma. Understanding more about the different factors affecting mental health among COVID-19 survivors can provide a basis for appropriate intervention to support our patients and to prevent detrimental effects in future pandemics.

## 2. Materials and Methods

### 2.1. Study Design and Respondents

This study recruited COVID-19 patients following their discharge from three centres, i.e., Hospital Sungai Buloh (a designated COVID-19 government hospital in Malaysia), Hospital Canselor Tuanku Muhriz (HCTM) (a public teaching hospital) and the Agro Exposition Park Serdang (MAEPS) Quarantine Centre.

Data were collected cross-sectionally at two different points, i.e., the first group at one month post-hospitalisation (between April 2020 and October 2020) and the second group at six months post-hospitalisation (between November 2020 and July 2021). Cases were recruited during spikes in COVID-19 cases in Malaysia during March 2020 and October 2020 until January 2021. Subjects were recruited via consecutive sampling during both study periods. The inclusion criteria included: (1) those who had been diagnosed with COVID-19 and had a positive COVID-19 Polymerase Chain Reaction (PCR) test, (2) those who were discharged from the three designated centres, (3) those who were 18 years old and above, and (4) those who were able to read and write in the English or Malay languages. Moreover, patients were excluded from the study if they were: (1) medically or mentally unstable or (2) non-Malaysians. All discharged COVID-19 patients who met all the study’s selection criteria were invited to participate in the study via electronic mail, phone call, or text message. Before signing an informed consent form, respondents who voluntarily agreed to participate were informed about the study’s procedures, purposes, participation benefits and risks, and assurance of anonymity, as well as their right to withdraw from the study at any point in time. Those who provided consent were asked to complete the questionnaires using an online survey platform, Google Forms. Data cleaning was performed to remove duplicate responses.

The sample size was estimated by using the G*Power 3.1.9.7 sample size calculator for estimating the sample size of two independent means, where type I error (α) = 0.05, Power (1 − β) = 0.8, allocation ratio = 1:1, and effect size = 0.35 (with reference to a study on the psychological impact of COVID-19 on COVID-19 survivors in China [32]). The estimated sample size needed was 244 subjects (inclusive of 20% dropout), whereby 122 subjects were needed per group/cohort.

### 2.2. Data Collection and Measures

Subjects were identified and recruited consecutively from the discharge registries of COVID-19 patients for the study period from the three centres. Selected patients were contacted and informed regarding the study. Respondents who consented to participate were divided according to their discharge dates into either the one-month or six-months post-hospitalisation groups. All respondents were administered a sociodemographic and clinical characteristics questionnaire, the Malay version of the Kessler Screening Scale for Psychological Distress (K6), and the Malay version of the Explanatory Model Interview Catalogue stigma scale (EMIC) during their assessments.

The outcome variable of this study was the level of psychological distress. It was measured using the Kessler Screening Scale for Psychological Distress (K6). The K6 assesses distress based on questions about anxiety and depressive symptoms that a person has experienced in the most recent 4-week period. It is a six-item self-rated psychological screening instrument developed by Kessler et al. (2002) [33]. The Malay version of the K6 was validated by Tiong et al. (2018) with good reliability and validity and exhibited good internal consistency with a Cronbach’s α of 0.859. It registered a sensitivity and specificity of 78.1% and 75.8%, respectively [34].

The Explanatory Model Interview Catalogue stigma scale (EMIC) is a 15-item self-rated scale originally designed to measure stigma among patients with leprosy. Higher scores indicate higher levels of perceived stigma. It was shown to be a valid and reliable tool in assessing stigma among recovered patients with COVID-19, with acceptable internal consistency with a Cronbach’s α of 0.79 [35]. The adaptation of the Malay version for the EMIC scale was performed by two independent bilingual native Malay-speaking language professionals, and the backwards translation into English language was carried out by another two bilingual native English-speaking language professionals who had not seen the original English version of the EMIC. Then, the translated and back-translated copies of the EMIC were examined by a panel of experts before a draft of the Malay version of the EMIC was constructed. The draft of the Malay version of the EMIC was administered to 20 COVID-19 survivors to screen for semantic quality, comprehensibility, redundancy of words and sentences, and duration of administration. Following feedback from the interviews of the 20 COVID-19 survivors, the semantic quality, comprehensibility, redundancy of words and sentences, and duration of administration were acceptable, and hence, no further amendments were made by the panel of experts. The Malay version of the EMIC was then administered to the COVID-19 survivors in this study for the validation study. The EMIC-SS-M reported an acceptable internal consistency with a Cronbach’s α of 0.727, and its domains reported an acceptable Cronbach’s α ranging from 0.708 to 0.795. EFA and CFA confirmed that the EMIC-SS-M consisted of 15 items in 4 domains [36].

Data on gender, age, ethnicity, employment and income, marital status, education status, previous medical or psychiatric illness, and counselling-seeking behaviour were recorded. The response to gender was reported in two groups: male or female. Age was recorded as a continuous variable. Ethnicity was recorded as either Malay or non-Malay. Employment status was recorded in three groups: retired, unemployed/housewife/students, and employed. Monthly income was reported in three groups: those who earned less than RM 5000/month, those who earned between RM 5000 to RM 10,000/month, and those who earned more than RM 10,000/month. Marital status was assessed in two groups: married and single/divorced/separated. Education status was reported in three groups: those who obtained up to primary, secondary, and tertiary education. Responses to history of medical and psychiatric illnesses and counselling seeking behaviour were recorded either as “Yes” or “No”.

### 2.3. Statistical Analysis

All data were analysed using the Statistical Package for Social Sciences, version 26 (SPSS 26; SPSS Inc., Chicago, IL, USA). Descriptive statistics for socio-demographic and clinical characteristics, as well as the total EMIC score and total K6 score, were reported. All categorical variables (gender, age, ethnicity, employment and income, marital status, education status, previous medical or psychiatric illness, and counselling seeking behaviour) were reported with a frequency and percentage. Moreover, all continuous variables (K6 and EMIC scores) were not normally distributed (*p* > 0.05 under the Kolgomorov–Smirnov test) and were thus reported with the median and interquartile range (IQR). The difference in sociodemographic and clinical characteristics between the one-month and post six-months post-hospitalisation groups of COVID-19 survivors were evaluated with Pearson’s chi-squared test, whereas the difference in total K6 and EMIC scores between the two groups were evaluated using the Mann–Whitney U test. The associations between socio-demographic and clinical variable, the EMIC scores (independent variables), and the K6 scores (dependent variable) in the two groups at different periods after discharge were measured using a multivariate general linear model with bootstrapping with 2000 replications. The statistical significance was *p* < 0.05, and all *p* values were two-sided.

### 2.4. Ethics

This study was approved by the Medical Research Committee of the Universiti Kebangsaan Malaysia Medical Centre (UKM PPI/111/8/JEP-2020-352) and the Medical Research and Ethics Committee of the Ministry of Health Malaysia (NMRR-20-1288-55105), and it abides by the regulations of the 1964 Declaration of Helsinki and its subsequent amendments.

## 3. Results

### 3.1. Respondent Characteristics

A total of 371 respondents were enrolled in the study, with 219 in the post one-month hospitalisation group and 152 in the post six-months hospitalisation group. All respondents’ socio-demographic and clinical characteristics are summarised in Table 1. In the context of socio-demographic characteristics, the two groups of COVID-19 survivors differed in employment status and monthly income. Those in the one-month post-hospitalisation group consisted of more of those who were unemployed/housewifes/students (39.7%) compared with more employed respondents in those in the six-months post-hospitalisation group (75.7%; *p* < 0.001). Those in the one-month post-hospitalisation group consisted of a larger proportion of people with a monthly income between RM 5000 to RM 10,000/month (84.0%) compared with a larger proportion of people with a monthly income of less than RM 5000 in those in the six-months post-hospitalisation group (62.5%; *p* < 0.001).

In terms of clinical characteristics, only counselling-seeking behaviour registered a significant difference between the two groups of COVID-19 survivors, whereby the proportion of respondents who sought counselling services was less in the one-month post-hospitalisation group (31.4%) compared with those in the six-months post-hospitalisation group (41.4%; *p* = 0.031). In addition, the level of perceived stigma was significantly higher in the one-month post-hospitalisation group (median = 11.0, IQR = 9.0) compared with that of the six-months post-hospitalisation group (median = 5.5, IQR = 8.0; *p* < 0.001). No significant difference in the level of psychological distress was noted between the two groups of COVID-19 survivors.

### 3.2. Associations among Demographic and Clinical Characteristics, Perceived Stigma, and Psychological Distress among COVID-19 Survivors One Month after Discharge

The associations between the demographic and clinical characteristics, perceived stigma, and severity of psychological distress among COVID-19 survivors one month after discharge are presented in Table 2. The multivariate general linear model indicated that employment status, monthly income, education status, history of psychiatric illnesses, and counselling-seeking behaviour were significantly associated with psychological distress among COVID-19 survivors after one month of being discharged from the hospital. It was found that significantly lower psychological distress was documented in those who retired (*B* = −2.207, 95% confidence interval [95% CI] = −4.139 to −0.068, *p* = 0.034) compared with those who were employed, in those who only received up to primary education (*B* = −2.474, 95% CI = −4.500 to −0.521, *p* = 0.014) compared with those with tertiary education, and those with an income of more than RM 10,000 per month (*B* = −1.576, 95% CI = −2.714 to −0.505, *p* = 0.006) compared with those with an income of RM 5000 to 10,000 per month, respectively. In contrast, those with a history of psychiatric illnesses (*B* = 6.363, 95% CI = 2.599 to 9.676, *p* = 0.002) compared to with those without a history, as well as those who sought counselling services (*B* = 1.737, 95% CI = 0.385 to 3.117, *p* = 0.016) compared with those who did not, had a significantly higher severity of psychological distress. Although a higher perceived stigma was documented in this group, it was not significantly associated with psychological distress (*p* = 0.487).

### 3.3. Associations among Demographic and Clinical Characteristics, Perceived Stigma, and Psychological Distress in COVID-19 Survivors Six Months after Discharge

At six-months after being discharged from the hospital, significantly higher psychological distress was documented in respondents who had a history of psychiatric illness (*B* = 2.887, CI = 0.469–6.437, *p* = 0.038) compared with those without a history and in those who exhibited counselling-seeking behaviour (*B* = 1.480, CI = 0.173–2.618, *p* = 0.032) compared with those who did not seek counselling services, respectively. In contrast, a higher degree of perceived stigma contributed to the greater severity of psychological distress among COVID-19 survivors after six months of discharge (*B* = 0.197, CI = 0.089–0.300, *p* = 0.002). The associations among socio-demographic and clinical characteristics, perceived stigma, and the severity of psychological distress among COVID-19 survivors six months after discharge from the hospital are illustrated in Table 3.

## 4. Discussion

This study compares the severity of psychological distress and the associations among socio-demographic and clinical characteristics, perceived stigma, and severity of psychological distress among COVID-19 survivors at one month and six months post-hospitalisation. We found that there was no difference in the severity of psychological distress among COVID-19 survivors between the one month and six months groups. Our findings depict that COVID-19 survivors may have had persistent and long-term distress even after they recovered from the acute infection. This finding is consistent with a study of COVID-19 survivors conducted in another country, whereby the authors found that, after 9 months of being infected by COVID-19, about 19% of patients still reported psychological distress [10].

However, different factors were found to be associated with psychological distress at these different time points after discharge. At one month after discharge, those with a higher income were associated with less distress. This is supported by other studies, which have reported that individuals with the lowest socio-economic status were the most vulnerable to psychological distress [37,38]. As the COVID-19 pandemic negatively impacted the economy through multiple restrictions that brought businesses and jobs to a standstill and possibilities of job retrenchments, those with a lower income were affected more, as they had less savings to prepare for these threats. Those with a higher income may have had more financial security in the face of these economic difficulties. This study also found that retired patients and those with primary school education had less distress at one month after discharge. There is accumulating evidence of retirement had positive effects on mental health, likely because of relief from stressful work, increased leisure time, increased physical activity, longer sleep duration, and fewer sleep difficulties [39]. Those with primary school education were found to have lower distress, which could be explained by lower levels of insight into the significance of emotional symptoms leading to less-reported psychological symptoms [40]. However, as denoted by our study, these socio-demographic characteristics (monthly income, employment status, and education level) were no longer protective against psychological distress after being labelled as COVID-19 survivors as time progressed. Hence, psychological or psychosocial factors may have played a role in maintaining the severity of psychological distress as time progressed after one was labelled as a COVID-19 survivor, as COVID-19 survivors in our study did not exhibit depreciation in the severity of psychological distress when we compared the COVID-19 survivors at one month and six months post-hospitalisation.

At both one and six months post-hospitalisation, higher psychological distress was reported in those with a history of psychiatric illness, and they exhibited counselling-seeking behaviour. This is supported by previous studies, which have pinpointed that the pandemic negatively affected those with mental illness, leading to a higher rate of relapse [41,42]. Psychological distress is a positive predictor of attitude towards seeking counseling, especially in those who were mental-health literate [43]. This finding is also echoed by another study in China among college students, which revealed that those who sought counselling had higher fear, depression, and trauma scores compared to those who did not seek counselling [29]. The experience of seeking psychological help and perceived mental health can effectively predict psychological help-seeking behaviour [29].

We found that the level of perceived stigma was higher in the earlier recovery period. This could be explained by insufficient knowledge and contradictory information about COVID-19, as it was a relatively new disease at that time. Stigma were also perpetuated by anxiety of getting infected and regarding the use of protective measures, with much misinformation in social media during the initial phase of the pandemic. The process of stigmatisation involves four main components, which are (1) labelling which personal characteristics are signaled or noticed as conveying important differences, e.g., being infected with COVID-19; (2) stereotyping which differences are linked to undesirable characteristics, e.g., those who were infected were not disciplined and not adherent to the preventive measures; thus, they were seen as having a bad attitude; (3) separating the normal group and the labelled group; and (4) discrimination of the labelled group through devaluation, rejection, and exclusion from the community [44]. In order to adequately combat stigma, Thornicroft identified three core problems where anti-stigma measures can be directed [45]. The core problems consist of problems of knowledge (ignorance), attitudes (prejudice), and behaviour (discrimination) [14]. Reductions in stigma over time could be due to continuous efforts by governmental and non-governmental bodies in the country and globally to provide correct information on COVID-19. This reduced the damaging effects of the infodemic during COVID-19 and increased empathy towards population groups at risk of stigmatisation. Our results also show that a persistent stigma contributed to psychological distress at the later phase of recovery. The association of a higher stigma with higher psychological distress has been consistently shown in multiple studies in the context of various medical conditions, such as schizophrenia, obesity, and infertility [45,46,47]. Despite the overall level of stigma being lower for those in the six-months post-hospitalisation group, a more persistent level of stigma contributed significantly to psychological distress in this period. Information overload regarding the dangers of COVID-19 may instill fear, which may also lead to a stigma. This may explain the persistent stigma despite many educational efforts regarding the infection.

The findings of our study should be interpreted in consideration of several limitations. Firstly, the respondents’ characteristics in our study were derived from three centres, not the whole of Malaysia’s COVID-19 patient population, therefore restricting the generalisability of our findings. Secondly, the demographic parameters of the two groups of respondents were not sufficiently matched, limiting the comparison between these groups. Third, our respondents were representative of two cross-sectional cohorts despite comparing COVID-19 survivors sampled at two different post-COVID-19 infection time points. As a result, the researchers were unable to confirm the associated factors reported in this study as causal. Furthermore, there were potential selection and response biases, as we only included those who were medically stable upon discharge from the hospital via an online data collection method which may have excluded patients who were not technologically savvy. Furthermore, data collection after six months may have led to recall bias, as it involved retrospective data and could be a negatively selected cohort that included more of those who were unemployed and those with counselling-seeking behaviour, as employed respondents may have been too busy to participate. Finally, we did not include other possible factors that may have contributed to psychological distress, such as coping methods and social support, which may be confounding factors in this study. There is still limited literature on assessing changes in stigma levels across time and their association with stress. Therefore, we recommend that future longitudinal cohort studies are conducted to confirm our findings, including assessments of coping strategies, social support, and changes in stigma over time with their association with stress.

Despite these limitations, this study is the first to determine the psychological distress factors, including stigma, among COVID-19 patients in Malaysia. Based on our findings, we included a few recommendations which could enhance the mental well-being of infected survivors during an infection pandemic. Because our findings indicate that a lower socioeconomic status may worsen psychological distress, financial assistance should be provided for those infected survivors who struggle with financial constraints. As those who were employed fared worse regarding mental health status compared with those who were retired, the government should emphasise screening for work-related stress among the working class and offer sufficient online counselling services to help them curb work-stress. Efficient and sufficient mental health services such as telemedicine (online mental health consultation during movement lockdown) and regular mental health follow-up services for those with a history of psychiatric illness should also be implemented to ensure that the mental health of this group of COVID-19 survivors is safeguarded. The perceived stigma among COVID-19 survivors should also be curbed effectively. In a systematic review of 24 studies on strategies for reductions in stigma on diseases, psychoeducation was the most common strategy and was effectively used to counter stigma [48].

## 5. Conclusions

From this study, we conclude that different factors may affect psychological distress at different periods of convalescence after a COVID-19 infection. COVID-19 does not discriminate against who it infects, and various factors, as highlighted in this study, change its mental health outcome, such as monthly income, employment status, education status, history of psychiatric illness, and counselling-seeking behaviour. Furthermore, persistent stigma contribute to psychological distress, and this suggests the need to apply psychosocial intervention to curb psychosocial issues among COVID-19 survivors, such as regular mental health screenings at primary care for those at risk and providing sufficient mental health services, such as rapid referrals to specialist centres, psychiatric consultations via face-to-face methods, or regular teleconsultation. This study provides valuable data for clinicians regarding the need to screen for perceived stigma among COVID-19 survivors, such as exploring how they view themselves with illnesses (i.e., low self-esteem or guilt) during follow-up consultations, and the need to provide psychosocial interventions, such as psychoeducation, to combat stigmatising perceptions. Social work assistance should be sought to curb factors that worsen psychological distress, such as financial status, work-related stress, those with a history of psychiatric illness, and the perceived stigma of being infected with COVID-19.

Currently, COVID-19 survivors in Malaysia undergo follow-up in their primary care clinic after discharge. Unfortunately, these consultations commonly focus on physical symptoms, whereas mental health symptoms appear to be overlooked. If detected, these symptoms are usually self-reported by patients instead of being detected by the clinician through active screening. Our findings not only emphasise the need to actively screen mental health symptoms but also suggest the need for interventions such as psychoeducation, psychiatric referrals, or telemedicine psychiatric consultations that many psychiatric centres provided following the COVID-19 pandemic.

## Figures and Tables

**Table 1 ijerph-20-03795-t001:** Sociodemographic and clinical characteristics of participants.

Variables	Subjects One Month after Discharge		Subjects Six Months after Discharge		*p*-Value
	Number of Participants (n = 219)	Percentage (%)	Number of Participants (n = 219)	Percentage (%)	
Age	32.0 ^#^	22.0 ^$^	34.0 ^#^	19.0 ^$^	0.329
Gender:					
Female	80	36.5	54	35.5	
Male	139	63.5	98	64.5	0.930
Ethnicity:					
Malay	168	76.7	125	82.2	
Non-Malay	51	23.3	27	17.8	0.248
Employment status:					
Retired	14	6.4	10	6.6	
Unemployed/housewife/student	87	39.7	27	17.8	
Employed	118	53.9	115	75.7	<0.001 *
Monthly household income:					
<RM 5000	0	0.0	95	62.5	
RM 5000–RM 10,000	184	84.0	43	28.3	
>RM 10,000	35	16.0	14	9.2	<0.001 *
Marital status:					
Married	98	55.3	83	54.6	
Single/divorced/separated	121	44.7	69	45.4	0.062
Education status:					
Primary education	9	4.1	8	5.3	
Secondary education	60	27.4	50	32.9	
Tertiary education	150	68.5	94	61.8	0.411
History of psychiatric illness:					
No	211	96.3	146	96.1	
Yes	8	3.7	6	3.9	0.884
History of medical illness:					
No	169	77.2	113	74.3	
Yes	50	22.8	39	25.7	0.531
Counselling-seeking behaviour:					
No	148	67.6	86	56.6	0.031 *
Yes	71	31.4	66	43.4	
Median total EMIC score	11.0 ^#^	9.0 ^$^	5.5 ^#^	8.0 ^$^	<0.001 *
Median total K6 score	9.0 ^#^	7.0 ^$^	10.0 ^#^	5.0 ^$^	0.272

* Statistical significance at *p* < 0.05, ^#^ = median, ^$^ = interquartile range.

**Table 2 ijerph-20-03795-t002:** Multivariate general regression model with bootstrapping with 2000 replications for socio-demographic and clinical characteristics, perceived stigma, and severity of psychological distress among COVID-19 patients after one month of being discharged from the hospital.

Variables	*B*	BCa 95% Confidence Interval		Standard Error	*p*-Value
		Lower	Upper		
Age	−0.033	−0.084	0.021	0.025	0.170
Gender:					
Male	Reference				
Female	0.629	−0.588	1.815	0.563	0.248
Ethnicity:					
Non-Malay	Reference				
Malay	−0.092	−1.351	1.274	0.631	0.874
Employment status:					
Employed	Reference				
Retired	−2.207	−4.139	−0.068	1.090	0.034 *
Unemployed/housewife/student	−0.751	−1.839	0.284	0.606	0.218
Monthly household income:					
RM 5000–RM 10,000	Reference				
>RM 10,000	−1.576	−2.714	−0.505	0.556	0.006 *
Marital status:					
Married	Reference				
Single/divorced/separated	0.326	−0.990	1.689	0.703	0.647
Education status:					
Tertiary education	Reference				
Secondary education	−0.516	−1.850	1.046	0.679	0.481
Primary education	−2.474	−4.500	−0.521	1.027	0.014 *
History of psychiatric illness:					
No	Reference				
Yes	6.363	2.599	9.676	1.865	0.002 *
History of medical illness:					
No	Reference				
Yes	0.127	−1.215	1.407	0.628	0.832
Counselling-seeking behaviour:					
No	Reference				
Yes	1.737	0.385	3.117	0.648	0.016 *
Total EMIC score	−0.028	−0.113	0.046	0.041	0.487

* Statistical significance at *p* < 0.05.

**Table 3 ijerph-20-03795-t003:** Multivariate general regression model with bootstrapping with 2000 replications for socio-demographic and clinical characteristics, perceived stigma, and severity of psychological distress among COVID-19 patients after six months of being discharged from the hospital.

Variables	*B*	BCa 95% Confidence Interval		Standard Error	*p*-Value
		Lower	Upper		
Age	−0.032	−0.126	0.060	0.037	0.395
Gender:					
Male	Reference				
Female	−0.079	−1.524	1.269	0.698	0.918
Ethnicity:					
Non-Malay	Reference				
Malay	−0.273	−2.269	1.769	0.964	0.808
Employment status:					
Employed	Reference				
Retired	0.071	−2.830	4.694	1.891	0.960
Unemployed/housewife/student	−0.610	−2.025	1.069	0.755	0.431
Monthly household income:					
<RM 5000	Reference				
RM 5000–RM 10,000	0.808	−1.799	3.053	1.390	0.549
>RM 10,000	0.451	−2.297	2.877	1.372	0.731
Marital status:					
Married	Reference				
Single/divorced/separated	0.326	−1.415	2.317	0.925	0.659
Education status:					
Tertiary education	Reference				
Secondary education	−0.516	−2.025	1.069	0.755	0.431
Primary education	−2.474	−2.830	4.694	1.891	0.960
History of psychiatric illness:					
No	Reference				
Yes	2.887	0.469	6.437	1.523	0.038 *
History of medical illness:					
No	Reference				
Yes	1.136	0.801	2.895	0.827	0.188
Counselling-seeking behaviour:					
No	Reference				
Yes	1.480	0.173	2.618	0.681	0.032 *
Total EMIC score	0.197	0.089	0.300	0.051	0.002 *

* Statistical significance at *p* < 0.05.

## Data Availability

The data presented in this study are available on request from the corresponding author. The data are not publicly available due to privacy and confidentiality concerns.

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
