# Peer review of "Stigma, Sociodemographic Factors, and Clinical Factors Associated with Psychological Distress among COVID-19 Survivors during the Convalescence Period: A Multi-Centre Study in Malaysia"

_ijerph, 2023, doi:10.3390/ijerph20053795_

Round 1

Reviewer 1 Report

Overall, the manuscript is in good shape. Few questions need to be addressed before publication. 
1) what is the significance of this study?

2) How the study sites were selected?

3) How the data were collected?

4) How about validation of study tool?

5) what are the golden standard to confirm psychological distress?

6) references from LMICs need to be added. 
7)Rephrase your conclusion and limitations.

Author Response

Author’s responses:

We deeply thank Reviewer 1 for the comments. Our responses are as follows:-

Q1: This study provides insight on how different factors and stigma may affect psychological distress at different periods of recovery after COVID-19 infection. It allows clinicians to screen COVID-19 patients at follow up and provide related interventions. These findings provide evidence for a more cost effective planning on support activities as we can target the specific factors for psychological distress at different period. From this study it was found that social factors affected psychological distress more during the early convalescence period thus clinician can focus on providing social work assistance to at the time of discharge. However persistent stigma led to psychological distress more at later part of recovery while social factors were not significant at this time. This shows the importance for a continued effort for anti-stigma related psychological interventions during follow up to prevent psychological distress. This is the first study in Malaysia that compares association of stigma and other factors at different times. As stigma is related to culture, it is vital for us to have this study conducted in the context of our local culture. As we could not deny possibility of recurrent pandemic in the future, it is important to be prepared to cope with its negative impact on mental health by studying our experiences during the COVID-19 pandemic.

Q2: The three study sites that participated in the study were selected as they were the main COVID-19 designated hospitals to receive COVID-19 inpatients at the time in this study were conducted.

MAEPS Quarantine centre - Was specifically opened to cater for COVID-19 patients who had minimal symptoms as all patients required to be in quarantine centre at that time. No home quarantine was allowed at the time of the study.

Hospital Sungai Buloh- Sungai Buloh Hospital is a public hospital which was converted to the  infectious disease centre that became the main facility and reference centre for treating COVID-19 in the country.

Hospital Canselor Tuanku Muhriz- HCTM is a university hospital where the main author and research team based from. It was also converted to one of the COVID-19 hospital during the pandemic.

Q3: Subjects were identified and recruited consecutively from the discharge registries of COVID-19 patients for the study period from the three centers. Selected patients were contacted and informed regarding the study. Respondents who consented to participate were divided according to their discharge dates into either the one month or six months post-hospitalization groups. All respondents were administered a sociodemographic and clinical characteristics questionnaire, the Malay version of the Kessler Screening Scale for Psychological Distress (K6), and the Malay version of the Explanatory Model Interview Catalogue stigma scale (EMIC) during their assessments.

Detailed information of the data collection methods were added in the manuscript, line  138 until 176.

Q4:Both study tools (K6 and EMIC) has been translated to the Malay versions and showed good internal consistency.

Detailed information of the validation of both questionnaires were added in the manuscript in line  177 until 205.

Q5: Kessler-6 questionnaire that were used in this study was a screening tool for psychological distress. In order to confirm and diagnose major psychiatric disorder, the gold standard to be used is the Mini International Neuropsychiatric Interview (MINI), based on the DSM 5 criteria. However the author did not proceed with MINI during the study. The participants who were screened as having high distress were referred for further assessments.

Q6: References from lower and middle income countries were added in line 58 and 70.

Q7: The conclusion and limitations were rephrased on page 11, starting from line 385 and page 12.

Reviewer 2 Report

Dear Authors,

congratulations on your research.

The title reflects well the content of the article, indicates the type of research. The article is relevant and considering the pandemic era and the impact that the pandemic brought, it is certainly current.

Abstract.

I suggest writing a structured abstract. Also, add numerical indicators of the most important results.

Introduction.

The introduction has good background, but it is too long (as many as 31 references) and very tiring to read. Be clear and specific, the introduction is too essay-like.

Methodology.

The sample and its design are described in detail, the criteria for inclusion and exclusion of patients are listed. The instrument and method of data collection are described, and the statistical analysis is clearly described. Considering the scope of the described work methodology, I suggest dividing the text into sub-chapters: Participants, Instrument, Statistical analysis, Ethics, etc.

The results.

The results are clearly presented, graphically and textually. The indicators can be clearly read from the tables. Discussion. The discussion is thoroughly written. It shows the main results and compares them with other similar works. The authors clearly pointed out the limitations of the study.

Conclusion.

In the conclusion, the authors state: "Furthermore, persistent stigma contributed to psychological distress and this suggested the need to apply psychosocial intervention to curb with psychosocial issue among COVID-19 survivors. This study provides valuable data for clinicians regarding the need to screen for perceived stigma among COVID-19 survivors and to provide psychosocial intervention and seek for social work assistance to curb with factors that worsen psychological distress, such as financial status, work-related stress, those with history of psychiatric illness, and perceived stigma of being infected with COVID- 19." I suggest that concrete implications for practice be clearly highlighted (what psychosocial intervention? how to apply the need to screen for perceived stigma? which psychosocial intervention? etc). How might the article's findings affect current clinical practice in Malaysia? I suggest the authors to be concrete with their conclusions.

Thank you, kind regards

Author Response

Author’s responses:

We thank Reviewer 2 for the comments and input. Our responses are as follows:-

In regards to the abstract, the author had added numerical indicators as advised. The abstract was written in accordance to IJERPH author guidelines where it was written in the style of structured abstracts, but without headings. The structure includes 1) Background and Aim, 2) Methods, 3) Results & 4)Conclusion. 

The author had shortened the introduction as advised.

The author had divided the text to subchapters as advised: 2.1. Study Design and Respondents, 2.2. Data Collection and Measures,  2.3. Statistical Analysis , & 2.4. Ethics  

In regards to the conclusion, the author has added the suggested information in line 431 until 452.

Reviewer 3 Report

This is a very well done pilote study to explore the effect of a COVID-19 infactions that causes a hospitalisation

*There is a possible bias in patient selection, try to mention that as a limoitation:

1/Voluntary collaboration after hospital discharge = positive selection of the  functionally better patients while patients with low performance status at that moment will not be included

2/On-line questionnaires exclude all patients that are not on-line, or elderly patients, of less educated patients...

 *After discharge  1 and 6 m: collection of data = retrospective data and that can be another limitation

 *The after 6m cohort is a negatively selected cohort of unemployed and counselling seeking behavior persons (maybe many" long covid" patients) because all patients that are employed again after 6m or no longer counseling seeking behavior will not collaborate because they are too busy professionally, another limitation of teh study 

*proposal for discussion: in future studies can it be interesting to follow the same patients 1m, 3m, 6m and 12m after hospital discharge to explore the stigma and stress over time

Author Response

Author’s response:

We thank Reviewer 3 for the uplifting comments and helpful input. The author has added the information above as advised in line 393-405.

Please see attachment. Thank you.

Round 2

Reviewer 1 Report

No further comments.